# *AaCaMKs* Positively Regulate Development, Infection Structure Differentiation and Pathogenicity in *Alternaria alternata*, Causal Agent of Pear Black Spot

**DOI:** 10.3390/ijms24021381

**Published:** 2023-01-10

**Authors:** Qianqian Jiang, Yongcai Li, Renyan Mao, Yang Bi, Yongxiang Liu, Miao Zhang, Rong Li, Yangyang Yang, Dov B. Prusky

**Affiliations:** 1College of Food Science and Engineering, Gansu Agricultural University, Lanzhou 730070, China; 2Institute of Postharvest and Food Sciences, The Volcani Center, Agricultural Research Organization, Rishon LeZion 7505101, Israel

**Keywords:** *Alternaria alternata*, *AaCaMKs*, development, infection structure differentiation, pathogenicity

## Abstract

Calcium/calmodulin-dependent protein kinase (CaMK), a key downstream target protein in the Ca^2+^ signaling pathway of eukaryotes, plays an important regulatory role in the growth, development and pathogenicity of plant fungi. Three AaCaMKs (AaCaMK1, AaCaMK2 and AaCaMK3) with conserved PKC_like superfamily domains, ATP binding sites and ACT sites have been cloned from *Alternaria alternata*, However, their regulatory mechanism in *A. alternata* remains unclear. In this study, the function of the *AaCaMKs* in the development, infection structure differentiation and pathogenicity of *A. alternata* was elucidated through targeted gene disruption. The single disruption of *AaCaMKs* had no impact on the vegetative growth and spore morphology but significantly influenced hyphae growth, sporulation, biomass accumulation and melanin biosynthesis. Further expression analysis revealed that the *AaCaMKs* were up-regulated during the infection structure differentiation of *A. alternata* on hydrophobic and pear wax substrates. In vitro and in vivo analysis further revealed that the deletion of a single *AaCaMKs* gene significantly reduced the *A. alternata* conidial germination, appressorium formation and infection hyphae formation. In addition, pharmacological analysis confirmed that the CaMK specific inhibitor, KN93, inhibited conidial germination and appressorium formation in *A. alternata*. Meanwhile, the *AaCaMKs* genes deficiency significantly reduced the *A. alternata* pathogenicity. These results demonstrate that *AaCaMKs* regulate the development, infection structure differentiation and pathogenicity of *A. alternata* and provide potential targets for new effective fungicides.

## 1. Introduction

*Alternaria alternata*, the causal agent of pear black spot, is a serious latent pathogen of mango [1], citrus [2], and pear [3], among other fruits, during their developmental period. It infects the fruits via the styles or peel during the growing season and remains latent until fruit maturity when it causes severe postharvest losses and food safety concerns. Pre-infection, *A. alternata* undergoes a series of processes to prepare for its infection, including forming a specialized infection structure necessary for its pathogenicity [4]. Precisely, the spores attach and germinate into the infective structure on the fruit surface, initiating the infection process. During this process, several signal pathways participate in external and internal signal transduction in response to the physical and chemical cues on the fruit surface. However, the growth patterns and development of *A. alternata* are more complex. Therefore, understanding the molecular mechanisms underlying host and *A. alternata* interactions is critical for effectively controlling the postharvest diseases caused by *A. alternata*.

The plant epidermis can protect the plants against pathogen infection; however, its hydrophobic [5], waxy [6], and other physicochemical cues also have inductive effects on pathogen infection. For example, in *Ustilago maydis*, the cuticle induces the hyphae and appressorium differentiation [7], while in *Magnaporthe oryzae*, the appressorium formation was induced on hydrophobic substrates [8]. In addition, the hydrophobic and pear wax substrates stimulate the differentiation of the *A. alternata* infection structure [9]. At the same time, *AaPKAc* and *AaPDEL/AaPDEH* in cAMP/PKA pathway [10,11], *AaPLCs* in Ca^2+^ signaling pathway [12], *AaHog1* in the MAPK cascades pathway [13], and transmembrane protein *AaSho1* [14] are involved in the differentiation of *A. alternata* infection structure under hydrophobic and pear wax substrates. However, how the pathogen responds to host epidermal signals until the initiation of the infection needs to be elucidated further.

Ca^2+^ is a common second messenger that mediates several physiological processes through the phosphorylate downstream target proteins calmodulin, *CaMK* and calcineurin [15]. The Ca^2+^ signal transduction pathway is one of the most important pathways in eukaryotes, involved in regulating various physiological processes, such as growth, spore germination, appressorium formation and pathogenicity [16,17,18].

*CaMK* is a key target protein in the Ca^2+^ signaling pathway, which regulates the growth, stress response and pathogenicity of pathogens [19,20]. For example, the deletion of *CaMK* in *Puccinia striiformis f.* sp. *tritici*, led to reduced vegetative growth and virulence loss. In the filamentous fungal pathogen *Neurospora crassa*, *CaMK* regulated the growth, thermo-tolerance and stress responses [21]. Similarly, *CaMK* regulates cell cycling, osmotic stressing and mating in *Saccharomyces cerevisiae* [22] and the growth, environmental stress tolerance and pathogenicity of *Arthrobotrys oligospora* [23]. However, the specific roles of *CaMK* in *A. alternata* development, infection structure differentiation and pathogenicity remain unknown.

Currently, three isotypes of calcium/calmodulin-dependent protein kinase, *AaCaMK1*, *AaCaMK2* and *AaCaMK3*, which are 1212, 1200 and 2349 bp long, respectively, have been cloned from *A. alternata* [24]. Further bioinformatics analysis revealed that the three CaMK contain the protein kinase catalytic-like superfamily domains (PKC_Like Superfamily). The conserved kinase domains of AaCaMK1 and AaCaMK2 belong to the catalytic domain of CaMK ser/thr protein kinase (STKc_CaMK), while that of AaCaMK3 belongs to the catalytic domain of liver kinase B1 (LKB1) and calmodulin-dependent protein kinase kinase (CaMKK) (STKc_LKB1_CaMKK). The homology analysis revealed that the homology of AaCaMK1, AaCaMK2 and AaCaMK3 with *Setosphaeria turcica* CAK1, CAK2 and CAK3 was as high as 94.32, 97.49 and 86.57%, respectively. Thus, we speculated that *AaCaMKs* might regulate the development and pathogenicity of *A. alternata.* In this study, the function of *AaCaMKs* in the development, differentiation of the infection structure and pathogenicity of *A. alternata* was elucidated through molecular biology and pharmacological methods. The results enhance our understanding of fungal biology, which is useful in developing better disease control strategies.

## 2. Results

### 2.1. AaCaMKs Are Not Essential for the Vegetative Growth of A. alternata but Are Indispensable for Sporulation and Biomass Accumulation

To assess the function of *AaCaMKs* in *A. alternata*, the gene deletion mutants, Δ*AaCaMK1*, Δ*AaCaMK2*, Δ*AaCaMK3* and complementary strains Δ*AaCaMK1*-C, Δ*AaCaMK2*-C and Δ*AaCaMK3*-C were obtained by homologous recombination and PEG-mediated protoplast transformation. The wild-type (WT), *ΔAaCaMKs* and the complementary strain *ΔAaCaMKs*-C spore and hyphae morphology on PDA at 3, 5 and 7 days of incubation revealed that the colony morphology and growth of these strains were comparable (Figure 1A,B). However, Δ*AaCaMK1* and Δ*AaCaMK3* mutants produced a large number of conidia, 2-fold higher than WT, whereas the conidia produced by Δ*AaCaMK2* mutant was 24% less than WT (Figure 2A). Compared to the WT strain, the single deletion *AaCaMKs* recorded a significantly reduced biomass accumulation after 5 d of incubation (Figure 2B). However, sporulation and biomass accumulation defects were rescued in *ΔAaCaMKs*-C strains. These results implied that the deletion of the *AaCaMKs* gene had no significant effect on the *A. alternata* growth; rather, it was involved in the sporulation and biomass accumulation.

### 2.2. AaCaMKs Regulate Hyphae Morphology Development in A. alternata

Observations under the scanning electron microscope revealed that the WT strain spores were elliptical and full of content and were nearly the same as those of the *ΔAaCaMKs* mutants (Figure 3A). In addition, the WT strain hyphae grew luxuriantly with many lateral branches but appeared short with abnormal branching in *ΔAaCaMK1* and *ΔAaCaMK3*. However, in the *ΔAaCaMK2* strain, the hyphae were not remarkably different from the WT strain (Figure 3B). These results imply that *AaCaMKs* played a key role in *A. alternata* hyphae growth.

### 2.3. Expression Analysis of AaCaMKs Gene during A. alternata Growth and Development

The expression levels of *AaCaMK2* and *AaCaMK3* were significantly lower than *AaCaMK1* in the WT, implying that under normal conditions, these three genes all play a role in *A. alternata* growth and development. However, *AaCaMK1* had a stronger regulatory role (Figure 4A). However, compared to the WT, the expression levels of *AaCaMK2* and *AaCaMK3* were dramatically up-regulated in *ΔAaCaMK1* (Figure 4B), suggesting that *AaCaMK2* and *AaCaMK3* together played complementary roles when *AaCaMK1* was knocked-out. Besides, *AaCaMK1* was up-regulated in *ΔAaCaMK2*, while *AaCaMK3* was down-regulated compared to the control (WT) (Figure 4C), implying that *AaCaMK1* played a major role in the *ΔAaCaMK2* mutant. The expression levels of *AaCaMK1* and *AaCaMK2* were up-regulated in *ΔAaCaMK3* compared to the WT (Figure 4D), implying that they both play a role in *ΔAaCaMK3* mutant. These results suggested that *AaCaMKs* may cooperatively regulate the growth and development of *A. alternata*.

### 2.4. AaCaMKs Are Crucial for the A. alternata Infection Structure Differentiation Induced on the Hydrophobic and Pear Wax Substrates

#### 2.4.1. AaCaMKs Are Up-Regulated during *A. alternata* Infection Structural Differentiation

The qRT-PCR analysis revealed that *AaCaMKs* were significantly up-regulated during the *A. alternata* infection structural differentiation in hydrophobic and pear wax substrates. However, *AaCaMK3* expression was lower than *AaCaMK1* and *AaCaMK2* (Figure 5A,B). On the hydrophobic substrate, *AaCaMKs* were significantly up-regulated at the appressorium (6 h) and infection hyphae formation stages (8 h) compared to the spore germination stage (2 h). Nevertheless, *AaCaMK1* and *AaCaMK2* reached the highest expression at the appressorium formation stage (6 h), with expressions 7 and 12-fold that of the control, respectively. However, *AaCaMK3* had the highest expression at the infection hyphae formation stage (8 h), which was 15-fold that of the control (Figure 5A). Similarly, compared to the spore germination stage (2 h), the expression level of *AaCaMKs* genes was significantly up-regulated at all stages of *A. alternata* infection structure differentiation under the pear wax-induced substrate. However, the expression of *AaCaMK1*, *AaCaMK2*, and *AaCaMK3* reached the highest levels during the germ tube elongation stage (4 h), with expression levels 5, 9 and 12 times that of spore germination stage (2 h), respectively (Figure 5B). Therefore, *AaCaMKs* were involved in the infection structural differentiation of *A. alternata* induced on hydrophobic and pear wax substrates.

#### 2.4.2. In Vitro Test

Spore germination and appressorium formation are key steps in fungal development and pathogenesis. The findings in this study revealed that spore germination and appressorium formation was significantly reduced in *AaCaMKs* deletion mutants with hydrophobic and pear wax substrates (*p* < 0.05). Precisely, appressorium formation in Δ*AaCaMK1*, Δ*AaCaMK2* and Δ*AaCaMK3* was significantly decreased to 14, 56 and 23% of WT under hydrophobic induction at 6 h post-inoculation, and 35, 56 and 64% on pear wax extract coated surface, respectively. However, the spore germination and appressorium formation of *ΔAaCaMKs*-C strains were almost recovered to the WT level (Figure 6).

#### 2.4.3. In Vivo Test

The appressorium and infective hyphae formation of WT, *ΔAaCaMKs* and *ΔAaCaMKs*-C were observed on the pear fruit peel. Compared to the WT, the appressorium formation on Δ*AaCaMK1*, Δ*AaCaMK2* and Δ*AaCaMK3* were reduced by 52, 45 and 51% at 6 h of induction, respectively (Figure 7A). After 12 h post-inoculation, the infection hyphae formation levels in Δ*AaCaMK1*, Δ*AaCaMK2* and Δ*AaCaMK3* were significantly reduced by 65, 74 and 65% compared to the WT, respectively (Figure 7B).

### 2.5. KN93 Inhibited A. alternata Spore Germination and Appressorium Formation

The KN93 treatment inhibited *A. alternata* conidial germination and appressorium formation in a dose-dependent manner. However, the appressorium formation was significantly inhibited (Figure 8A,B). Precisely, the *A. alternata* spore germination was 43% lower than the control 2 h after being treated with 40 µM KN93 (Figure 8A). Furthermore, adding 20 µM KN93 into the *A. alternata* conidial suspension drastically decreased the appressorium formation to approximately 84% after 4 h incubation at 28 °C on a hydrophobic substrate. At the same time, treatment with 40 µM KN93 completely inhibited the appressorium formation (Figure 8B). Additionally, the spore germination and appressorium formation on hydrophobic and pear wax substrates treated with KN93 were consistently lower than for the untreated substrates (Figure 8C,D). At 8 h post-treatment, the appressorium formation on KN93 treated hydrophobic and pear wax substrates was significantly reduced by 53 and 30%, respectively (Figure 8D). These results further confirmed that *AaCaMKs* regulated the *A. alternata* infection structural differentiation.

### 2.6. AaCaMKs Regulate A. alternata Virulence

The analysis of *AaCaMKs* role in the *A. alternata* pathogenicity using infection assays on ‘Zaosu’ pear fruit revealed that the WT, *ΔAaCaMKs* and *ΔAaCaMKs*-C *A. alternata* caused typical black spots on pears 3 d post-inoculation. As shown in Figure 9, *ΔAaCaMKs* exhibited a 56~61% reduction in lesion diameters on pear fruit 5 d after inoculation. However, the pear fruit lesion area caused by *ΔAaCaMKs*-C was equal to that of the WT. These results implied that *AaCaMKs* were required for *A. alternata* pathogenicity.

### 2.7. AaCaMKs Regulate the Melanin Content of A. alternata

The intracellular melanin content of Δ*AaCaMK1* and Δ*AaCaMK3* were significantly higher than WT. While the intracellular melanin content in Δ*AaCaMK2* was not significantly changed, the melanin contents of Δ*AaCaMK1* and Δ*AaCaMK3* were significantly increased by 39 and 75%, respectively, compared to the WT (Figure 10A). In contrast, the extracellular melanin content of *ΔAaCaMKs* was significantly lower than WT. Precisely, the melanin content of Δ*AaCaMK1* and Δ*AaCaMK2* was decreased by 47 and 59%, respectively (Figure 10B). These results imply that *AaCaMKs* regulate the melanin content in *A. alternata*.

## 3. Discussion

*CaMK* is a serine/threonine protein kinase that plays significant and conserved roles in regulating growth, conidial germination, appressorium formation, stress response and pathogenicity in many fungal pathogens. For example, *CaMK* regulates conidial germination, appressorium formation, melanin production and pathogenicity in *M. oryzae* [25,26]. In addition, the deletion of the *CaMK* gene in *Aspergillus nidulans* caused a defect in nuclear division and spores germination; thus, the disruption of *CMKB* is lethal [27]. Recent reports have also revealed that *Cmk2* has an additional function of calcium tolerance in budding yeast [28].

In this study, three single deletion mutants (*ΔAaCaMK1, ΔAaCaMK2* and *ΔAaCaMK3*) and complementary strains (*ΔAaCaMK1*-C, *ΔAaCaMK2*-C and *ΔAaCaMK3*-C) of *AaCaMKs* were constructed from the WT *A. alternata*. Phenotype analysis revealed that the targeted deletion of *AaCaMKs* affected sporulation and biomass accumulation. In contrast, the single deletion of *AaCaMKs* had no significant impact on *A. alternata* vegetative growth and spore morphology compared to the WT. Similarly, the deletion of the *CaMK* gene resulted in significant sporulation defects, and the *CpkB* mutant did not affect the vegetative growth of *S. nodorum* [29]. In *M. oryzae*, *CaMK* deletion mutants also showed strong growth defects and produced reduced conidia [30]. These results indicate that the roles of *CaMK* vary among the different plant fungal pathogens. At the same time, the expression analysis revealed that the deletion of a single *AaCaMK* increases the expression of the other two genes to complement the defects. Alternatively, *AaCaMKs* could have functional redundancy in regulating *A. alternata* growth and development, or the *AaCaMKs* positively or negatively regulate each other. This could be a direct or indirect regulation. Similarly, Kumar et al. [21] found that *camk-1* regulates the growth and development of *N. crassa* using a gene deletion mutant, although there might be a *camk-1* substitute gene.

The plant epidermis can effectively prevent water transpiration and reduce mechanical damage and insect infection. It also plays an important role in pathogen recognition and infection structure formation [31,32,33]. Several studies have revealed that the hydrophobic and pear wax substrates induce appressorium differentiation of plant pathogenic fungi [7,9,31,34,35]. Herein, the expression level of *AaCaMK1*, *AaCaMK2* and *AaCaMK3* were significantly up-regulated during the *A.alternata* infection structural differentiation, implying that *AaCaMKs* is involved in the differentiation of the *A. alternata* infection structure induced by hydrophobic and pear wax substrates. These results are similar to the previous report in *S. turcica*, where the expression levels of *CaMK* were significantly up-regulated during the infection structural differentiation [20]. In addition, the *PsCaMKL1* expression was up-regulated at the early infection stages of *Puccinia striiformis f. sp. tritici* [36]. However, the deletion of *ΔAaCaMKs* caused a considerable reduction in spore germination and appressorium formation based on the in vivo and in vitro tests, which suggests a potential role of *AaCaMKs* in regulating the differentiation of the infection structure of *A. alternata*. This is consistent with the findings of Hu et al. [26], who reported that the *CaMK* gene in *M. oryzae* was involved in regulating spore germination and appressorium formation. Liu et al. [19] also demonstrated delayed conidial germination and appressorial formation in the *MoCMK1* deletion mutant of *M. oryzae*. KN93, a specific inhibitor of CaMK, also significantly inhibited *A. alternata* spore germination and appressorium formation in a dose-dependent manner. Precisely, the spore germination and appressorium formation of *A. alternata* treated with KN93 were significantly inhibited by hydrophobic and pear wax substrates, similar to the findings in previous studies using *M. oryzae*, *Bipolarismayd* and *Puccinia striiformis f.* sp. *Tritici* [26,37,38].

Generally, melanin is interconnected with the development and pathogenicity of fungal pathogens [39,40]. Interestingly, we found that *AaCaMKs* could affect melanin biosynthesis in *A. alternata*, but not the colony color. Similarly, the *CaMK* disturbance inhibits melanin biosynthesis in *C. gloeosporioides* [41]. However, the specific regulatory mechanism of melanin biosynthesis by *CaMK* is unclear. A thorough analysis of *A. alternata* inoculation on wounded pear fruit revealed that *AaCaMKs* were correlated with *A. alternata* pathogenicity. Hu et al. [26] also reported that *CaMK* regulates pathogenicity in *M. oryzae*. Similarly, functional studies on *CpkB* and *CpkC* genes have demonstrated that *CaMK* plays significant roles in *Stagonospora nodorum* pathogenicity [29]. However, the specific regulation mechanism still needs to be studied further by constructing *AaCaMKs* gene double deletion and triple deletion mutants.

## 4. Materials and Methods

### 4.1. Fungal Strain, Vector, Reagents and Fruits

The wild type (WT) *A. alternata* strain JT-03 was isolated from infected pear fruit and cultured on potato dextrose agar (PDA) at 28 °C for 5 d. The pCAMBIA1300-HPH and pC-NEO-NGFP vectors were provided by the Chinese Academy of Sciences. The pear wax used in the experiments was extracted from pears as described by Yin et al. [42] and then prepared as a 0.1% wax solution. KN93 and Gelbond PAG film were purchased from TopScience (Shanghai, China) and Univ-bio (Shanghai, China). The pear (*Pyrus bretchneideri* ‘Zaosu’) was harvested from a Tiaoshan Farm in Jingtai County, Gansu Province, China.

### 4.2. Construction of AaCaMKs Deletion Mutants and Complementation

The *AaCaMKs* knock-out vector was constructed based on a homologous recombination strategy (Appendix A). Two homologous recombination sequences (5′ and 3′ flank) flanking the target gene were amplified on the WT *A. alternata* strain using the primer pair *AaCaMK1/2/3*-up and *AaCaMK1/2/3*-down (Appendix A) and then inserted at the multiple cloning sites on pCHPH upstream and downstream of *hph*, respectively. Next, the plasmid vectors were transferred into the WT strain by PEG-mediated protoplast transformation [43]. The transformants were selected based on their growth on PDA containing 0.08 g L^−1^ hygromycin B, followed by screening and confirmation by PCR and qRT-PCR (Appendix A).

Subsequently, the *AaCaMKs* complementation strains (*ΔAaCaMKs*-C) were constructed following the method highlighted by Chen et al. [44]. Next, the complementation vectors were constructed by amplifying the cDNA of *AaCaMKs* and transformed into the *ΔAaCaMKs* mutants by PEG-mediated protoplast transformation. Successful complementation was then screened with G418 (0.25 g L^−1^). and the complementary strain *ΔAaCaMKs*-C was verified by PCR. The primers used are listed in Appendix A.

### 4.3. Growth and Development Assays

#### 4.3.1. Determination of Vegetative Growth, Sporulation and Biomass of *A. alternata*

*A. alternata* growth and conidiation were detected as previously described [10]. Precisely, the PDA medium was inoculated with 2 µL spore suspensions (1 × 10^5^ spores mL^−1^) of WT *A. alternata*, *ΔAaCaMKs* and *ΔAaCaMKs*-C strains and incubated at 28 °C. Their growth was examined at 3, 5 and 7 d, respectively.

For the sporulation assay, the conidia were collected and resuspended in 10 mL of ddH_2_O, then filtered to remove the hyphae and impurities after incubating at 28 °C for 3 d. The number of conidia collected was counted under a microscope using a hemocytometer. For the biomass estimation assay, the spore suspensions were inoculated on PDA, covered with sterile cellophane sheets and incubated at 28 °C for 5 d, after which the mycelia and cellophane sheets were removed and weighed. The experiments were replicated thrice.

#### 4.3.2. Spore and Hyphae Morphology

The spore morphology was assessed according to the method described by Hu et al. [45]. The WT and *ΔAaCaMKs* spore suspensions were centrifuged (5000× *g*, 5 min), then washed in three changes of PBS, and fixed in 2.5% glutaraldehyde for 3 h. Next, the spores were dehydrated by graded concentrations of ethanol (30, 50, 70, 80, 90 and 95%), and then isoamyl acetate was added. The spore morphology was observed under scanning electron microscope (SEM) (JSM-5600LV) at 3000 magnification. The hyphae morphology was characterized as described by Jimdjio et al. [46]. Briefly, 2 µL of WT and *ΔAaCaMKs* spore suspension were inoculated on PDA, covered with sterile cellophane, and then incubated at 28 °C for 3 d. The hyphae growing at the edge of the cellophane were cut and placed on a slide, then observed under a electron microscope and photographed.

### 4.4. Real-Time Quantitative Reverse Polymerase Chain (qRT-PCR) Reaction

The spore suspensions were inoculated on the hydrophobic and pear wax-coated hydrophobic film, incubated at 28 °C for 2, 4, 6 and 8 h and then harvested. Next, the total RNA was extracted from *A. alternata* JT-03 using the TRNzol reagent (TIANGEN, Beijing, China), and 2 µg of the extracted RNA was reverse transcribed to cDNA for qRT-PCR (Takara, Dalian, China). The expression level of the target gene on the respective days was calculated according to Livak and Schmittgen [47]. *GAPDH* was used as the reference gene. The primers used during qRT-PCR are listed in Appendix A.

### 4.5. Infection Structure Formation Assays

WT, *ΔAaCaMKs* and *ΔAaCaMKs*-C spore suspensions (20 µL each) were dripped onto hydrophobic and pear wax-coated hydrophobic film with three replicates and incubated at 28 °C. The percentage spore germination and appressorium formation were calculated under a microscope at 2, 4, 6 and 8 h after incubation. To further demonstrate the effect of *AaCaMKs* on the *A. alternata* infection structure formation in vivo, pear fruits were cut into small pieces (3 × 3 × 1 cm), and 20 µL of WT, *ΔAaCaMKs* and *ΔAaCaMKs*-C spores suspension were inoculated on pear peel surface. The subsequent method was referred to by Tang et al. [9]. All assays were performed in triplicate.

### 4.6. Pharmacological Test

KN93, a specific inhibitor of *CaMK*, was dissolved in dimethyl sulfoxide (DMSO). Next, *A. alternata* spores were suspended in KN93 at concentrations of 0 (CK), 10, 20, 30 and 40 µM, and 20 µL of each spore suspension was inoculated on the hydrophobic film and incubated at 28 °C. Subsequently, observations were made under the microscope every 2 h for 8 h. In addition, 20 µL of each spore suspension was inoculated onto hydrophobic and pear wax substrates and cultured at 28 °C, followed by observation under the microscope at 2, 4, 6 and 8 h post-inoculation.

### 4.7. Melanin Extraction and Measurement

Melanin was extracted and estimated following the method described by Gao et al. [48] in potato dextrose broth (PDB) incubated at 28 °C with shaking for 6 d. Briefly, the mycelia and filtrates were separated using four layers of gauze. Next, the filtrates were adjusted to pH 2 and centrifuged (8000× *g*, 30 min) to obtain the precipitates consisting of extracellular melanin. The mycelium obtained by filtration was then dried, and 0.25 g was accurately weighed and boiled in 30 mL of 1 M NaOH for 5 h. Subsequently, the mycelia were filtered, and the filtrate was adjusted to pH 2 and centrifuged (8000× *g*, 30 min) to obtain the intracellular melanin (precipitate).

Next, 5 mL of 7 M HCl was added to the precipitate and boiled for 2 h, then centrifuged (8000× *g*, 30 min), and the supernatant was discarded. The precipitate was then dissolved in 1 M NaOH and adjusted to pH 2 using 7 M HCl, while the supernatant was discarded after centrifugation (8000× *g*, 30 min). Finally, the precipitate was dissolved and fixed in 1 M NaOH, and its absorbance was measured at 400 nm using a UV spectrophotometer, using 1 M NaOH as a blank control. The results were presented in standard curve: y = x + 0.111/0.791 (x: the absorbance value of the sample measured at 400 nm; y: melanin content).

### 4.8. Pathogenicity Assays

For the pear infection assay, the pear fruits were disinfected using 1% sodium hypochlorite for 2 min, and three wounds were uniformly inflicted on the equator of each pear using sterile punching nails (3 mm in diameter and 5 mm in depth). Next, 20 µL of WT, *ΔAaCaMKs* and *ΔAaCaMKs*-C spore suspensions were inoculated into each wound. The pear fruits were then placed in plastic boxes and stored at room temperature. The lesion diameter was measured at 3, 5, 7 and 9 d after inoculation. Each treatment had nine fruits replicated three times.

### 4.9. Statistical Analysis

All assays in this study were replicated at least three times. Origin 8.5 was used for mapping. The average and standard error (±SE) were calculated in Microsoft Excel 2016. The statistical analysis was performed using SPSS 18.0. The differences between the means were compared using Duncan’s multiple range test at *p* < 0.05.

## 5. Conclusions

Three single deletion mutants and complementary strains of *AaCaMKs* were successfully constructed, and the functions of *AaCaMKs* in growth, development, infection structure differentiation and pathogenicity of *A. alternata* were elucidated. Our findings demonstrate that *AaCaMKs* are essential for sporulation, biomass accumulation, hyphae growth, melanin biosynthesis, infection structure differentiation and pathogenicity of *A. alternata*, but are not required for the growth and spore morphology. Therefore, *AaCaMKs* play diverse and essential roles in *A. alternata*. These results will widen our knowledge of the molecular mechanisms of disease progression caused by *A. alternata*, and provide potential drug targets for developing new effective fungicides.

## Figures and Tables

**Figure 1 ijms-24-01381-f001:**
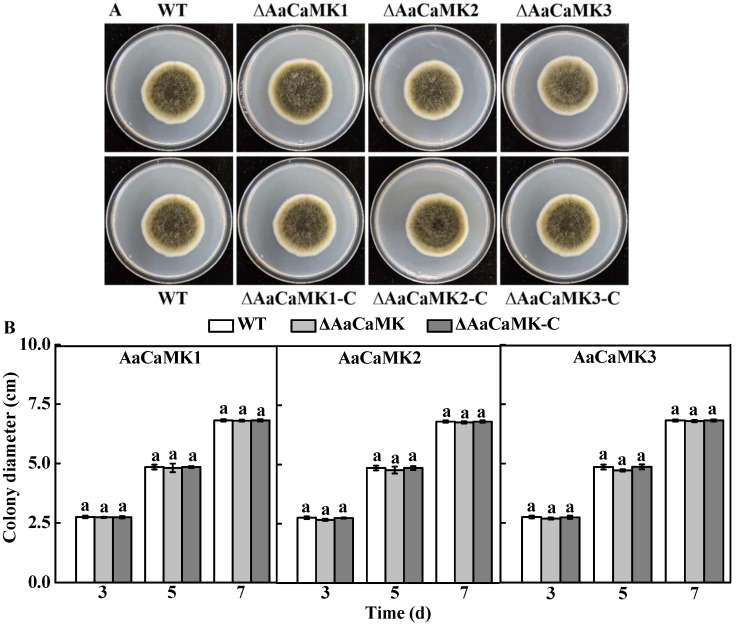
The colony morphology (**A**) and growth rates (**B**) of *A. alternata* on PDA medium 5 d after incubation at 28 °C. Vertical lines indicate the standard error (±SE) of the means. Different letters indicate significant differences (*p* < 0.05).

**Figure 2 ijms-24-01381-f002:**
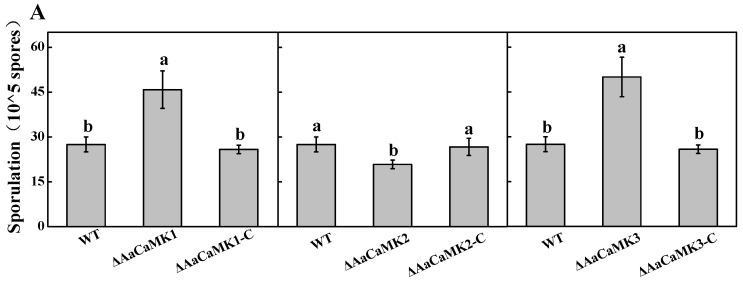
The sporulation rate (**A**) and biomass (**B**) of *A. alternata* on PDA medium 5 d after incubation at 28 °C. Bars indicate standard error (±SE). Different letters indicate significant differences (*p* < 0.05).

**Figure 3 ijms-24-01381-f003:**
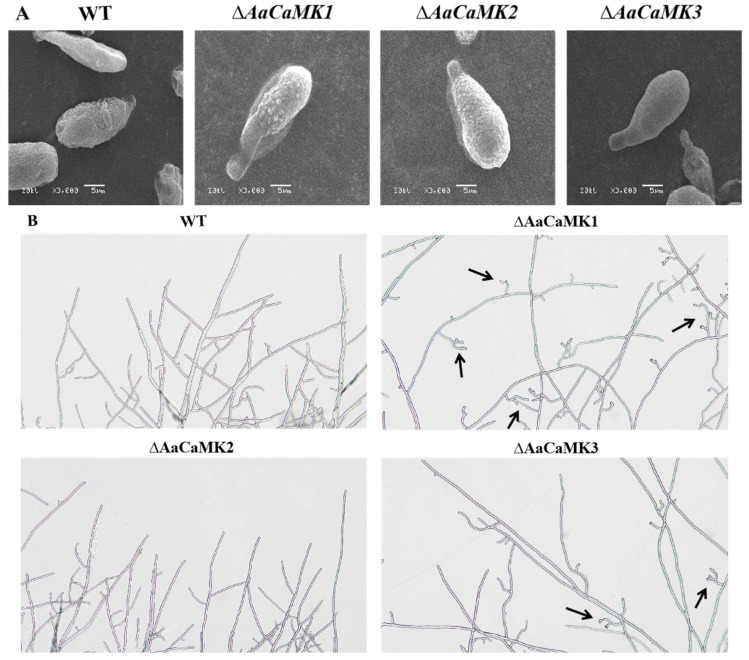
The spore (**A**) and hyphae morphology (**B**) of *A. alternata*. The black arrow indicates the abnormal branching hyphae.

**Figure 4 ijms-24-01381-f004:**
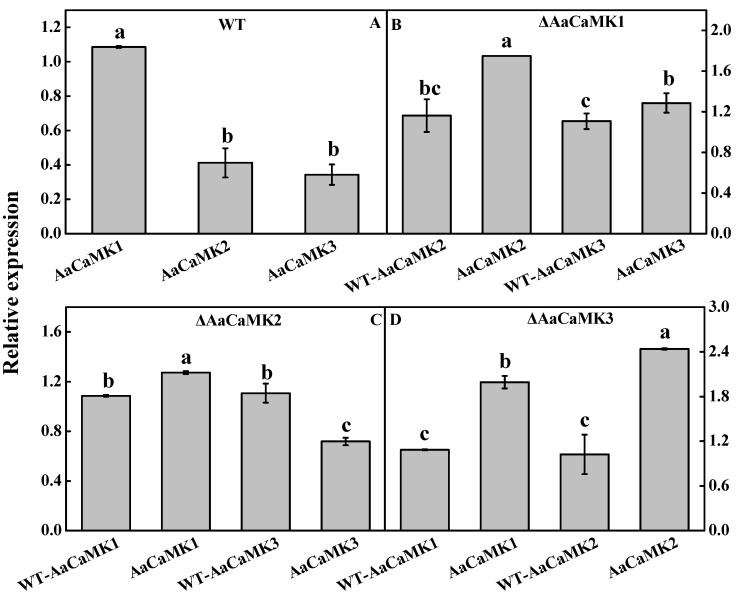
The transcription levels of *AaCaMKs* genes. The expression level of the *AaCaMKs* gene in WT strains was plotted as a control. (**A**) Expression levels of *AaCaMK1*, *AaCaMK2* and *AaCaMK3* in WT. (**B**) Expression levels of *AaCaMK2* and *AaCaMK3* in *ΔAaCaMK1*. (**C**) Expression levels of *AaCaMK1* and *AaCaMK3* in *ΔAaCaMK2*. (**D**) Expression levels of *AaCaMK1* and *AaCaMK2* in *ΔAaCaMK3*. Bars indicate standard error (±SE). Different letters indicate significant differences (*p* < 0.05).

**Figure 5 ijms-24-01381-f005:**
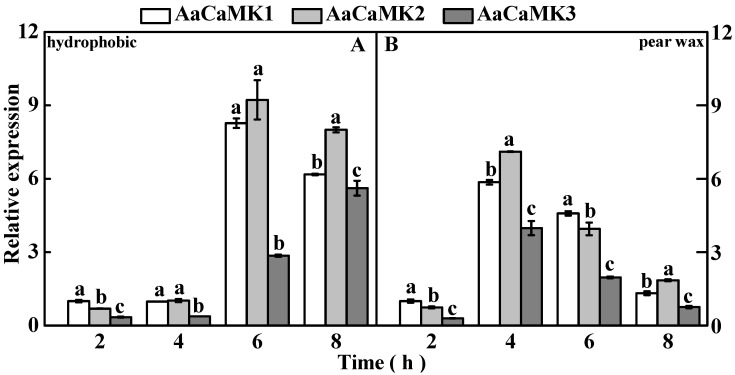
Relative expression levels of *AaCaMKs* during the infection structure differentiation stage of *A. alternata* on hydrophobic (**A**) and pear wax (**B**) substrates by qRT-PCR. Vertical lines indicate the standard error (±SE) of the means. Different letters indicate significant differences (*p* < 0.05).

**Figure 6 ijms-24-01381-f006:**
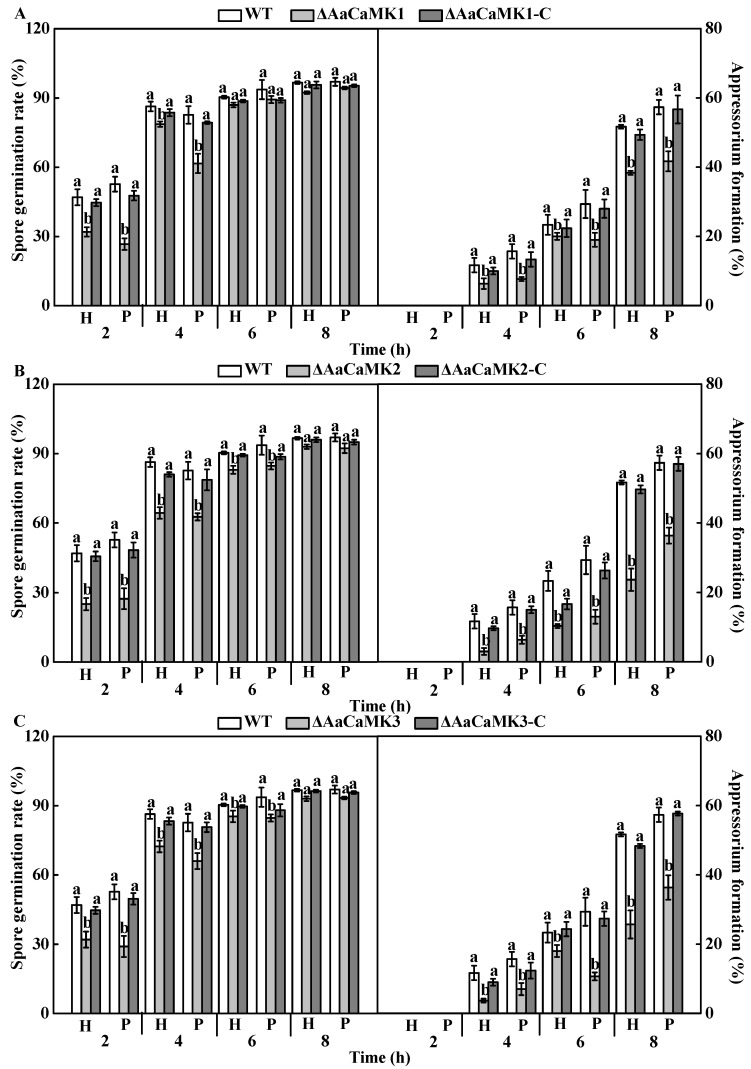
The spore germination and appressorium formation of *AaCaMK1* (**A**), *AaCaMK2* (**B**) and *AaCaMK3* (**C**) induced by hydrophobic (H) and pear wax (P). Bars indicate standard error (±SE). Different letters indicate significant differences (*p* < 0.05).

**Figure 7 ijms-24-01381-f007:**
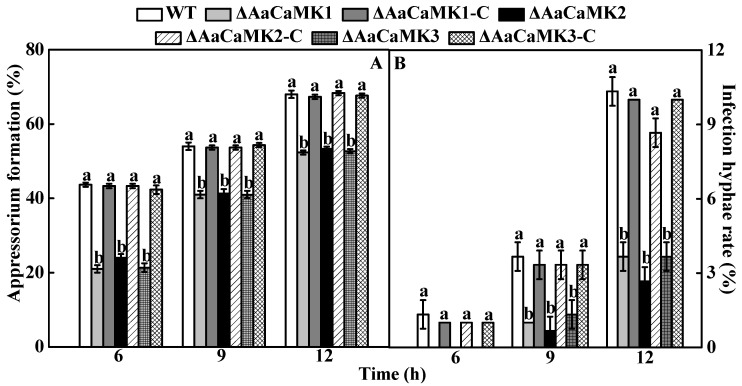
The *A. alternata* appressorium (**A**) and infection hyphae (**B**) formation on intact pear epidermis. Vertical lines indicate standard error (±SE). Different letters indicate significant differences (*p* < 0.05).

**Figure 8 ijms-24-01381-f008:**
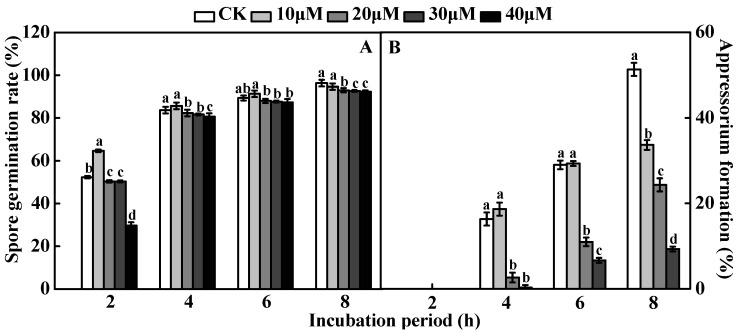
*A. alternata* spore germination and appressorium formation following treatment with KN93. (**A**) *A. alternata* spore germination at different KN93 concentrations; (**B**) *A. alternata* appressorium formation at different KN93 concentrations; (**C**) *A. alternata* spore germination following treatment with KN93 under hydrophobic (H) and pear wax (P) substrates; (**D**) *A. alternata* appressorium formation following treatment with KN93 using hydrophobic (H) and pear wax (P) substrates. The ddH_2_O was used as a control. The vertical line indicates a standard error (±SE). Different letters indicate significant differences (*p* < 0.05).

**Figure 9 ijms-24-01381-f009:**
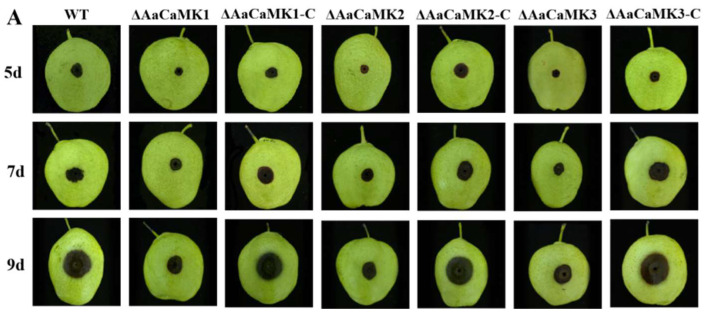
The disease progression (**A**) and lesion diameter (**B**). Means and standard deviations were calculated from three replicates. Different letters in the graph indicate statistical differences (*p* < 0.05).

**Figure 10 ijms-24-01381-f010:**
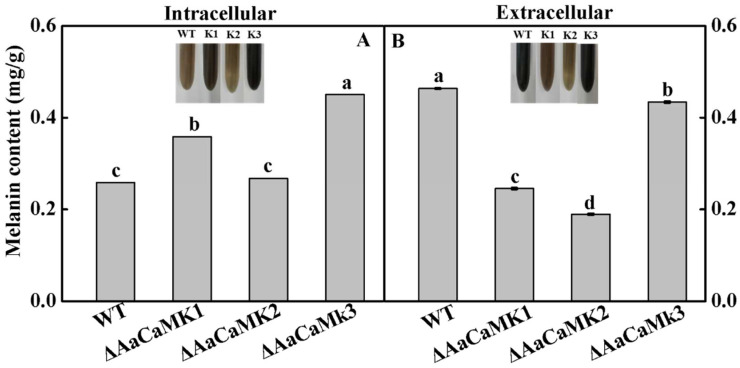
*A. alternata* melanin content. (**A**) The intracellular melanin content in WT, Δ*AaCaMK1,* Δ*AaCaMK2* and Δ*AaCaMK3.* (**B**) The extracellular melanin content in WT, Δ*AaCaMK1,* Δ*AaCaMK2* and Δ*AaCaMK3.* Bars represent standard deviations of means of three replicates. Different letters in the graph indicate statistical differences (*p* < 0.05).

## Data Availability

The datasets generated and analyzed during the current study are available from the corresponding author on reasonable request.

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
