# Peer review of "AaCaMKs Positively Regulate Development, Infection Structure Differentiation and Pathogenicity in Alternaria alternata, Causal Agent of Pear Black Spot"

_ijms, 2023, doi:10.3390/ijms24021381_

Round 1
Reviewer 1 Report
The manuscript entitled "AaCaMKs positively regulate development, infection structure differentiation and pathogenicity in Alternaria alternata, causal agent of pear black spot" described the role of CaMKs in A. alternata. The authors have successfully constructed three single deletion mutants and complementary strains of AaCaMKs, and elucidated the role of AaCaMKs in sporulation, biomass, hyphae growth, melanin biosynthesis, infection structure differentiation and pathogenicity of A. alternata, but not required for growth and spore morphology. The followings are the questions that I am concerned about in this manuscript:
The authors did not perform the sequence homology and source analysis of CaMK. It is not clear whether the protein encoded by these genesindeed have the desired biochemical function. The authors also did not determine the amount of CaMK. Is there three in other species as well? How conservative are they? What causes the phenotypic variations between each mutant? Also, the evidence for CaMK's influence on A. alternata was insufficient, the authors ought to look at the phenotypic alterations following knockout all the three genes simultaneously.
Despite the significant scientific insights based on the experimental data, the narrative interpretations are problematic at times and often cause contradiction and confusion. In addition to the above problems, there are some grammatical errors in the manuscript, including but not limited to the following:
L15: Add a"," after "however".
L64: "deletion of CaMK reduced vegetative growth and loss of virulence." should change to " deletion of CaMK led to vegetative growth reduced and virulence loss."
L83: The author should describe where the genescame from, how to screen for it, and also the conserved role.
L84:The "Δ" of "ΔAaCaMK1" shouldn’t be Italic.
L87: "complemented" should be " complementary".
L88: The "d" of "After 3, 5 and 7 d of culture" should be given the full name where it first appeared.
L95: The authors ought to look at the phenotypic alterations following knockout all the three genessimultaneously, which can further clarify the effects of CaMKs on A. alternata.
L111-113: "ΔAaCaMK1 and ΔAaCaMK3 mutants appeared short and abnormal branching hyphae, and ΔAaCaMK1 was more than ΔAaCaMK3 strain." This sentence should to be modified because it's unclear.
L122-135: This portion of the evidence is insufficient since the expression of these three genes cannot demonstrate that they may cooperate to control growth and development. Have the authors attempted to complemented each of these three genes back into the mutants in order to determine if the defects can be restored?
L208: Data of mutants should be added in Figure 8to better reflect the role of AaCaMKs.
L233-239: What is the difference between the extracellular melanin and intracellular melanin content on pathogenicity?
L246-247: Confusion remains as to which of the two "M. grisea" and "M. oryzae" are used, as both are now used by different authors. But now "M. oryzae" is in mainstream use.
L255-268: How well conserved are AaCaMKsacross different organisms? Do other species also have three CamKs? Do they also play a redundant role?
L285: "M. grisea" should change to "M. oryzae"
Author Response
Dear editor,
Thanks for you and reviewer comments concerning our manuscript entitled “AaCaMKs positively regulate development, infection structure differentiation and pathogenicity in Alternaria alternata, causal agent of pear black spot” (Manuscript ID: ijms-2091557). Those comments are very valuable for revising and improving our paper. We have studied all the comments carefully and have made the necessary revision as required. The revision is marked in blue in our revised manuscript. Attached please find the revised version, and responses to reviewers’ specific comments are detailed below:
Reviewer: 1
The manuscript entitled "AaCaMKs positively regulate development, infection structure differentiation and pathogenicity in Alternaria alternata, causal agent of pear black spot" described the role of CaMKs in A. alternata. The authors have successfully constructed three single deletion mutants and complementary strains of AaCaMKs, and elucidated the role of AaCaMKs in sporulation, biomass, hyphae growth, melanin biosynthesis, infection structure differentiation and pathogenicity of A. alternata, but not required for growth and spore morphology. The followings are the questions that I am concerned about in this manuscript:
The authors did not perform the sequence homology and source analysis of CaMK. It is not clear whether the protein encoded by these genes indeed have the desired biochemical function. The authors also did not determine the amount of CaMK. Is there three in other species as well? How conservative are they? What causes the phenotypic variations between each mutant? Also, the evidence for CaMK's influence on A. alternata was insufficient, the authors ought to look at the phenotypic alterations following knockout all the three genes simultaneously.
Response: Thank you very much for your professional suggestions, cloning, bioinformatics AaCaMKs in Alternaria alternata have been studied and published (Jiang et al., 2021 in Microbiology China). The results showed that three isotypes of calcium/calmodulin-dependent protein kinase designated AaCaMK1, AaCaMK2 and AaCaMK3 identified in A. alternata were 1212, 1200 and 2349 bp length respectively, the bioinformatics analysis revealed that these three CaMK all contain the PKC_like superfamily domains (PKC_Like Superfamily). The conserved kinase domains of both AaCaMK1 and AaCaMK2 belonged to the catalytic domain of CaMK ser/thr protein kinase (STKc_CaMK) while that of AaCaMK3 belonged to the catalytic domain of liver kinase B1 (LKB1) and calmodulin dependent protein kinase kinase (CaMKK) (STKc_LKB1_CaMKK). The homology analysis showed that the homology of AaCaMK1, AaCaMK2 and AaCaMK3 with Setosphaeria turcica CAK1, CAK2 and CAK3 were as high as 94.32%, 97.49% and 86.57%, respectively. We have added this reference in manuscript.
Thank you very much for your suggestion, all the three genes simultaneously knockout are underway in our laboratory for further elucidating synergy mechanism of AaCaMKs.
Despite the significant scientific insights based on the experimental data, the narrative interpretations are problematic at times and often cause contradiction and confusion. In addition to the above problems, there are some grammatical errors in the manuscript, including but not limited to the following:
L15: Add a"," after "however".
Response: Thank you for your carefully reviewing, the "," has been added in Line 15.
L64: "deletion of CaMK reduced vegetative growth and loss of virulence." should change to " deletion of CaMK led to vegetative growth reduced and virulence loss."
Response: Thank you for your carefully reviewing, the sentence has been changed in Line 65-67.
L83: The author should describe where the genes came from, how to screen for it, and also the conserved role.
Response: Thank you very much for your suggestion, cloning, bioinformatics AaCaMKs in Alternaria alternata have been studied and published (Jiang et al., 2021 in Microbiology China). we have added this reference and related characteristics of AaCaMKs have been added in Introduction section in Line73-82.
L84:The "Δ" of "ΔAaCaMK1" shouldn’t be Italic.
Response: Thank you for your carefully reviewing, the "Δ" of "ΔAaCaMK1" has been changed to Italic in Line 94.
L87: "complemented" should be " complementary".
Response: Thank you for your carefully reviewing, the "complemented" has been changed to "complementary" in Line 97.
L88: The "d" of "After 3, 5 and 7 d of culture" should be given the full name where it first appeared.
Response: Thank you for your carefully reviewing, the "d" has been changed to "days" in Line 98.
L95: The authors ought to look at the phenotypic alterations following knockout all the three genes simultaneously, which can further clarify the effects of CaMKs on A. alternata.
Response: Thank you very much for your suggestion, all the three genes simultaneously knockout are underway in our laboratory for further elucidating synergy mechanism of AaCaMKs.
L111-113: "ΔAaCaMK1 and ΔAaCaMK3 mutants appeared short and abnormal branching hyphae, and ΔAaCaMK1 was more than ΔAaCaMK3 strain." This sentence should to be modified because it's unclear.
Response: Thank you for your carefully reviewing, we have changed this sentence in Line 120-122.
L122-135: This portion of the evidence is insufficient since the expression of these three genes cannot demonstrate that they may cooperate to control growth and development. Have the authors attempted to complemented in order to determine if the defects can be restored?
Response: Thank you very much for your professional suggestions, the experiments of all the three genes simultaneously knockout and each of these three genes back into the mutants are underway in our laboratory for further elucidating synergy mechanism of AaCaMKs.
L208: Data of mutants should be added in Figure 8 to better reflect the role of AaCaMKs.
Response: Thank you very much for your professional suggestions, the results in Figure 8 has been used to confirm the role of CaMK on the spore germination and appressorium formation of A. alternata at the protein level using specific inhibitors of CaMK KN93.
L233-239: What is the difference between the extracellular melanin and intracellular melanin content on pathogenicity?
Response: Our previous results showed that melanin was a key factor affecting the pathogenicity of A. alternata, so the extracellular melanin and intracellular melanin content were detected respectively to investigate the difference. However, the specific pathogenic mechanism of melanin or extracellular melanin and intracellular melanin in A. alternata need further study.
L246-247: Confusion remains as to which of the two "M. grisea" and "M. oryzae" are used, as both are now used by different authors. But now "M. oryzae" is in mainstream use.
Response: Thank you for your carefully reviewing, "M. grisea" in the whole article has been modified.
L255-268: How well conserved are AaCaMKs across different organisms? Do other species also have three CaMKs? Do they also play a redundant role?
Response: CaMKs across different organisms has been analyzed in our previous article (Jiang et al., 2021 in Microbiology China) and also described in Induction section. At present, similar reports of functional redundancy of CaMKs have only been found in N. crassa.
L285: "M. grisea" should change to "M. oryzae"
Response: Thank you for your carefully reviewing, "M. grisea" has been changed to "M. oryzae".

Reviewer 2 Report
Jiang et al. constructed three single deletion mutant and complementary strains of Alternaria alternata (causing black spot of pear) in order to investigate the role of the AaCaMK genes in vegetative growth, development, morphology of infection structures and pathogenicity of the fungus. The experimental setup was well designed and the obtained results are sound. Therefore, the study makes an important contribution to the understanding of the factors leading to disease development of A. alternata. However, the presentation of the manuscript could be significantly improved.
The manuscript would benefit from a thorough revision of the English language. In particular, the introduction is difficult to follow due to a number of imprecise statements or incomplete sentences. Furthermore, the introduction should better introduce the reader into the topic, e.g. by providing a short overview about the CaMK genes so far know in A. alternata. At the end of the introduction, the authors should clearly define the hypotheses of their study. The materials and methods chapter should be checked in order to avoid bullet point style sentences such as in lines 393/394.
In Figure 4, the same scaling should be used for all the y-axes related to the relative gene expression in order to make the comparison of data easier. The same would apply for Figure 8, the y-axis regarding appressorium formation.
In subchapter 4.3.2., the title does not match the content and is the same as the one of the previous subchapter.
The supplementary tables and figures should be removed from the main manuscript and placed in a separate document.
Author Response
Dear editor,
Thanks for you and reviewer comments concerning our manuscript entitled “AaCaMKs positively regulate development, infection structure differentiation and pathogenicity in Alternaria alternata, causal agent of pear black spot” (Manuscript ID: ijms-2091557). Those comments are very valuable for revising and improving our paper. We have studied all the comments carefully and have made the necessary revision as required. The revision is marked in blue in our revised manuscript. Attached please find the revised version, and responses to reviewers’ specific comments are detailed below:
Reviewer: 2
Jiang et al. constructed three single deletion mutant and complementary strains of Alternaria alternata (causing black spot of pear) in order to investigate the role of the AaCaMK genes in vegetative growth, development, morphology of infection structures and pathogenicity of the fungus. The experimental setup was well designed and the obtained results are sound. Therefore, the study makes an important contribution to the understanding of the factors leading to disease development of A. alternata. However, the presentation of the manuscript could be significantly improved.
The manuscript would benefit from a thorough revision of the English language. In particular, the introduction is difficult to follow due to a number of imprecise statements or incomplete sentences.
Response: As your suggestion, we totally agree and the manuscript has been revised and re-polished by a native English speakers.
Furthermore, the introduction should better introduce the reader into the topic, e.g. by providing a short overview about the CaMK genes so far know in A. alternata. At the end of the introduction, the authors should clearly define .
Response: As your suggestion, bioinformatics AaCaMKs in Alternaria alternata have been described in Introduction section which has been published (Jiang et al., 2021 in Microbiology China). The hypotheses of this study have also added at the end of the introduction.
The materials and methods chapter should be checked in order to avoid bullet point style sentences such as in lines 393/394.
Response: As your suggestion, we have checked and revised.
In Figure 4, the same scaling should be used for all the y-axes related to the relative gene expression in order to make the comparison of data easier. The same would apply for Figure 8, the y-axis regarding appressorium formation.
Response: Thank you very much for your suggestion, in Figure 4, there are different controls in different strains in each Figure, so different y-axes have been used in different strains/Figures.
In Figure 8, spore germination and appressorium formation are two different indicators, and there is no direct correlation between them, so different y-axis have been used.
In subchapter 4.3.2., the title does not match the content and is the same as the one of the previous subchapter.
Response: Thank you for your carefully reviewing, the title has been changed to “ 4.3.2 Spore and hyphae morphology observation” in Line 348.
The supplementary tables and figures should be removed from the main manuscript and placed in a separate document.
Response: The supplementary tables and figures has been removed from the main manuscript and placed in a separate document.
